# Tackling Neural Architecture Search
# With Quality Diversity Optimization

**Lennart Schneider**[1]  **Florian Pfisterer**[1]  **Paul Kent**[2]  **Juergen Branke**[3]  **Bernd Bischl**[1]
**Janek Thomas**[1]

[1]Department of Statistics, LMU Munich, Germany
[2]Mathematics of Real World Systems, University of Warwick, UK
[3]Warwick Business School, University of Warwick, UK

**Abstract**  Neural architecture search (NAS) has been studied extensively and has grown to become a research field with substantial impact. While classical single-objective NAS searches for the architecture with the best performance, multi-objective NAS considers multiple objectives that should be optimized simultaneously, e.g., minimizing resource usage along the validation error. Although considerable progress has been made in the field of multi-objective NAS, we argue that there is some discrepancy between the actual optimization problem of practical interest and the optimization problem that multi-objective NAS tries to solve. We resolve this discrepancy by formulating the multi-objective NAS problem as a quality diversity optimization (QDO) problem and introduce three quality diversity NAS optimizers (two of them belonging to the group of multifidelity optimizers), which search for high-performing yet diverse architectures that are optimal for application-specific niches, e.g., hardware constraints. By comparing these optimizers to their multi-objective counterparts, we demonstrate that quality diversity NAS in general outperforms multi-objective NAS with respect to quality of solutions and efficiency. We further show how applications and future NAS research can thrive on QDO.

## 1 Introduction

The goal of neural architecture search (NAS) is to automate the manual process of designing optimal neural network architectures. Traditionally, NAS is formulated as a single-objective optimization problem with the goal of finding an architecture that has minimal validation error [13, 35, 45, 47, 46, 63]. Considerations for additional objectives such as efficiency have led to the formulation of constraint NAS methods that enforce efficiency thresholds [1] as well as multi-objective NAS methods [10, 12, 37, 53, 36] that yield a Pareto optimal set of architectures. However,

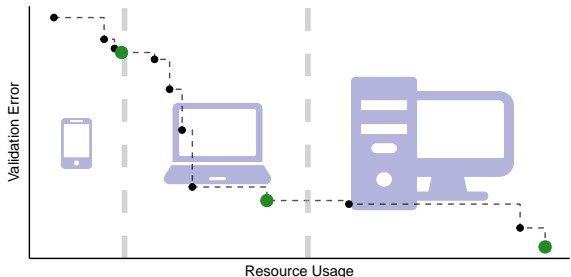

Figure 1: Optimizing neural network architectures for a discrete set of devices. We are interested in the best solution (green) within the constraints of the respective device (dashed vertical lines). Multi-objective optimization, in contrast, approximates the full Pareto front (black).

in most practical applications, we are not interested in the complete Pareto optimal set. Instead, we would like to obtain solutions for a discrete set of scenarios (e.g., end-user devices), which we henceforth refer to as *niches* in this paper. This is illustrated in Figure 1. A concrete example is finding neural architectures for microcontrollers [32] and other edge devices [38], e.g., in $\mu$NAS [32] architectures for "mid-tier" IoT devices are searched. To evaluate the benefits for larger devices, the search would need to be restarted with adapted constraints, thus wasting computational resources. Formulating the search as a multi-objective problem would also waste resources; once an architecture satisfies the constraints of a device, we are not interested in additional trade-offs, and we select only based on the validation error.

We therefore argue that the multi-objective NAS problem *can* and usually *should* be formulated as a *quality diversity optimization* (QDO) problem, which directly corresponds to the actual optimization problem of interest. The main contributions of this paper are: We (1) formulate multi-objective NAS as a QDO problem; (2) show how to adapt black-box optimization algorithms for the QDO setting; (3) modify existing QDO algorithms for the NAS setting; (4) propose novel multifidelity QDO algorithms for NAS; and (5) illustrate that our approach can be used to extend a broad range of NAS methods from conventional to Once-for-All methods.

## 2 Theoretical Background and Related Work

Let $\mathcal{A}$ denote a search space of architectures and $\Lambda$ the search space of additional hyperparameters controlling the training of an architecture $A$. Furthermore, let $f_{\text{err}} : \mathcal{A} \times \Lambda \to \mathbb{R}$ denote the validation error obtained after training an architecture $A \in \mathcal{A}$ with a set of hyperparameters $\lambda \in \Lambda$ for a given number of epochs ($\lambda_{\text{epoch}} \in \lambda$). Typically, we consider $\lambda \in \Lambda$ to be fixed, except for $\lambda_{\text{epoch}}$ in multifidelity methods, and we therefore omit $\lambda$ in the following. The goal of single-objective NAS is to find the architecture with the lowest validation error, $A^* := \arg\min_{A \in \mathcal{A}} f_{\text{err}}(A)$.

NAS methods can be categorized along three dimensions: search space, search strategy, and performance estimation strategy [13]. For chain-structured neural networks (simply a connected sequence of layers), cell-based search spaces have gained popularity [46, 45]. In cell-based search spaces, different kinds of cells – typically, a normal cell preserving dimensionality of the input and a reduction cell reducing spatial dimension – are stacked in a predefined arrangement to form a final architecture. Regarding search strategy, popular methods utilize Bayesian optimization (BO) [3, 8, 39, 24, 57], evolutionary methods [41, 34, 47, 46, 12], reinforcement learning [63, 64], or gradient-based algorithms [35, 45]. For performance estimation, popular approaches leverage lower fidelity estimates [31, 14, 64] or make use of learning curve extrapolation [8, 27].

**Multi-Objective Neural Architecture Search** Contrary to the single-objective NAS formulation, multi-objective NAS does not solely aim for minimizing the validation error but simultaneously optimizes multiple objectives. These objectives typically take resource consumption – such as memory requirements, energy usage or latency – into account [10, 12, 37, 53, 36]. Denote by $f_1, \ldots, f_k$ the $k \geq 2$ objectives of interest, where typically $f_1 = f_{\text{err}}$ and denote by $\mathbf{f}(A)$ the vector of objective function values obtained for architecture $A \in \mathcal{A}$, $\mathbf{f}(A) = (f_1(A), \ldots, f_k(A))'$. The optimization problem of multi-objective NAS is then formulated as $\min_{A \in \mathcal{A}} \mathbf{f}(A)$. There is no architecture that minimizes all objectives at the same time since these are typically in competition with each other. Rather, there are multiple Pareto optimal architectures reflecting different trade-offs in objectives approximating the true (unknown) Pareto front. An architecture $A$ is said to dominate another architecture $A'$ iff $\forall i \in \{1, \ldots, k\} : f_i(A) \leq f_i(A') \land \exists j \in \{1, \ldots, k\} : f_j(A) < f_j(A')$.

**Constrained and Hardware-Aware Neural Architecture Search** In contrast, *Constrained NAS* [62, 15, 55] solves the problem of finding an architecture that optimizes one objective (e.g., validation error) with constraints on secondary objectives (e.g., model size). Constraints can be naturally given by the target hardware that a model should be deployed on. *Hardware-Aware NAS* in turn searches for an architecture that trades off primary objectives [60] against secondary, hardware-specific

metrics. In Once-for-All [5], a large supernet is trained which can be efficiently searched for subnets that, e.g., meet latency constraints of target devices. For a recent survey, we refer to [1].

**Quality Diversity Optimization** The goal of a QDO algorithm is to find a set of high-performing, yet behaviorally diverse, solutions. Similarly to multi-objective optimization, there is no single best solution. However, whereas multi-objective optimization aims for the simultaneous minimization of multiple objectives, QDO minimizes a single-objective function with respect to diversity defined on one or more *feature functions*. A feature function measures a quality of interest and a combination of feature values points to a niche, i.e., a region in *feature space*. QDO could be considered a *set* of constrained optimisation problems over the same input domain where the niche boundaries are constraints in feature space. The key difference is that constrained optimisation seeks a single optimal configuration given some constraints, while QDO attempts to identify the optimal configuration for each of a set of constrained regions simultaneously. In this sense, QDO could be framed as a so-called multi-task optimization problem [43] where each task is to find the best solution belonging to a particular niche.

QDO algorithms maintain an archive of niche-optimal observations, i.e., a best-performing observed solution for each niche. Observations with similar feature values compete to be selected for the archive, and the solution set gradually improves during the optimization process. Once the optimization budget has been spent, QDO algorithms typically return this archive as their solution. QDO is motivated by applications where a group of diverse solutions is beneficial, such as the training of robot movement where a repertoire of behaviours must be learned [7], developing game playing agents with diverse strategies [44], and in automatic design where QDO can be used by human designers to search a large dimensional search space for diverse solutions before the optimization is finished by hand. Work on automatic design tasks have been varied and include air-foil design [18], computer game level design [16], and architectural design [9]. Recently, QDO algorithms were used for illuminating the interpretability and resource usage of machine learning models while minimizing their generalization error [52].

In the earliest examples, Novelty Search (NS; [29]) asks whether diversity alone can produce a good set of solutions. Despite not actively pursuing objective performance, NS performed surprisingly well in some settings and was followed by Novelty Search with Local Competition [30], the first true quality diversity (QD) algorithm. MAP-Elites [42], a standard evolutionary QDO algorithm, partitions the feature space a-priori into niches and attempts to identify the optimal solution in each of these niches. QDO has seen much work in recent years and a variant based on BO, BOP-Elites, was proposed recently [25]. BOP-Elites models the objective and feature functions with surrogate models and implements an acquisition function over a structured archive to achieve high sample efficiency even in the case of black-box features.

## 3 Formulating Neural Architecture Search as a Quality Diversity Optimization Problem

In the example in Figure 1, a quality diversity NAS (subsequently abbreviated as qdNAS) problem is given by the validation error and three behavioral niches (corresponding to different devices) that are defined via resource usage measured by a single feature function. Let $f_1 : \mathcal{A} \to \mathbb{R}, A \mapsto f_1(A)$ denote the objective function of interest (in our context, $f_{\text{err}}$). Denote by $f_i : \mathcal{A} \to \mathbb{R}, A \mapsto f_i(A), i \in \{2, \ldots, k\}, k \geq 2$ the feature function(s) of interest (e.g., memory usage). Behavioral niches $N_j \subseteq \mathcal{A}, j \in \{1, \ldots, c\}, c \geq 1$ are sets of architectures characterized via niche-specific boundaries $\mathbf{b}_{ij} = [l_{ij}, u_{ij}) \subseteq \mathbb{R}$ on the images of the feature functions. An architecture $A$ belongs to niche $N_j$ if its values with respect to the feature functions lie between the respective boundaries, i.e.:

$$A \in N_j \iff \forall i \in \{2, \ldots, k\} : f_i(A) \in \mathbf{b}_{ij}.$$

The goal of a QDO algorithm is then to find for each behavioral niche $N_j$ the architecture that minimizes the objective function $f_1$:

$$A_j^* := \arg\min_{A \in N_j} f_1(A).$$

In other words, the goal is to obtain a set of architectures $\mathcal{S} := \{A_1^*, \ldots, A_c^*\}$ that are diverse with respect to the feature functions and yet high-performing.

**A Remark about Niches** In the classical QDO literature, niches are typically constructed to be pairwise disjoint, i.e., a configuration can only belong to a single niche (or none) [42, 25]. However, depending on the concrete application, relaxing this constraint and allowing for *overlap* can be beneficial. For example, in our context, an architecture that fits on a mid-tier device should also be considered for deployment on a higher-tier device, i.e., in Figure 1, the boundaries indicated by vertical dashed lines resemble the respective upper bound of a niche whereas the lower bound is unconstrained. This results in niches being nested within each other, i.e., $N_1 \subsetneq N_2 \subsetneq \ldots \subsetneq N_c \subseteq \mathcal{A}$, with $N_1$ being the most restrictive niche, followed by $N_2$. In Supplement A, we further discuss different ways of constructing niches in the context of NAS.

### 3.1 Quality Diversity Optimizers for Neural Architecture Search

As the majority of NAS optimizers are iterative, we first demonstrate how any iterative optimizer can in principle be turned into a QD optimizer. Based on this correspondence, we introduce three novel QD optimizers for NAS: BOP-Elites*, qdHB and BOP-ElitesHB. Let $f_1 : \mathcal{A} \to \mathbb{R}, A \mapsto f_1(x)$ denote the objective function that should be minimized. In each iteration, an iterative optimizer proposes a new configuration (e.g., architecture) for evaluation, evaluates this configuration, potentially updates the incumbent (best configuration evaluated so far) if better performance has been observed, and updates its archive. For generic pseudo code, see Supplement B.

Moving to a QDO problem, there are now feature functions $f_i : \mathcal{A} \to \mathbb{R}, A \mapsto f_i(A), i \in \{2, \ldots, k\}, k \geq 2$, and niches $N_j, j \in \{1, \ldots, c\}, c \geq 1$, defined via their niche-specific boundaries $\mathbf{b}_{ij} = [l_{ij}, u_{ij}) \subseteq \mathbb{R}$ on the images of the feature functions. Any iterative single-objective optimizer must then keep track of the best incumbent per niche (often referred to as an *elite* in the QDO literature) and essentially becomes a QD optimizer (see Algorithm 1). The challenge

---

**Algorithm 1**: Generic pseudo code for an iterative quality diversity optimizer.

**Input** : $f_1, f_i, i \in \{2, \ldots, k\}, k \geq 2, N_j, j \in \{1, \ldots, c\}, c \geq 1, \mathcal{D}_{\text{design}}, n_{\text{total}}$
**Result**: $S = \{A_1^*, \ldots, A_c^*\}$

1  $\mathcal{D} \leftarrow \mathcal{D}_{\text{design}}$
2  **for** $j \leftarrow 1$ **to** $c$ **do**
3  $\quad$ $A_j^* \leftarrow \arg\min_{A \in \mathcal{D}_{|N_j}} f_1(A)$ # initial incumbent of niche $N_j$ based on archive
4  **end**
5  **for** $n \leftarrow 1$ **to** $n_{\text{total}}$ **do**
6  $\quad$ Propose a new candidate $A^\star$ # subroutine
7  $\quad$ Evaluate $y \leftarrow f_1(A^\star), \forall i \in \{2, \ldots, k\} : z_i \leftarrow f_i(A^\star)$
8  $\quad$ **if** $A^\star \in N_j \wedge y < f_1(A_j^*)$ **then**
9  $\quad\quad$ $A_j^* \leftarrow A^\star$ # update incumbent of niche $N_j$
10 $\quad$ **end**
11 $\quad$ $\mathcal{D} \leftarrow \mathcal{D} \cup \{(A^\star, y, z_2, \ldots, z_k)\}$
12 **end**

---

in designing an efficient and well-performing QD optimizer now mostly lies in proposing a new candidate for evaluation that considers improvement over all niches.

**Bayesian Optimization** A recently proposed model-based QD optimizer, BOP-Elites [25], extends BO [54, 17] to QDO. BOP-Elites relies on Gaussian process surrogate models for the objective function and all feature functions. New candidates for evaluation are selected by a novel acquisition function – the expected joint improvement of elites (EJIE), which measures the expected improvement to the ensemble problem of identifying the best solution in every niche:

$$\alpha_{\text{EJIE}}(A) := \sum_{j=1}^{c} \mathcal{P}(A \in N_j|\mathcal{D})\mathbb{E}_y\left[\mathbb{I}_{|N_j}(A)\right]. \tag{1}$$

Here, $\mathcal{P}(A \in N_j|\mathcal{D})$ is the posterior probability of $A$ belonging to niche $N_j$, and $\mathbb{E}_y\left[\mathbb{I}_{|N_j}(A)\right]$ is the expected improvement (EI; [23]) with respect to niche $N_j$:

$$\mathbb{E}_y\left[\mathbb{I}_{|N_j}(A)\right] = \mathbb{E}_y\left[\max\left(f_{\min_{N_j}} - y, 0\right)\right],$$

where $f_{\min_{N_j}}$ is the best observed objective function value in niche $N_j$ so far, and $y$ is the surrogate model prediction for $A$. A new candidate is then proposed by maximizing the EJIE, i.e., Line 6 in Algorithm 1 looks like the following: $A^\star \leftarrow \arg\max_{A \in \mathcal{A}} \alpha_{\text{EJIE}}(A)$.

**BOP-Elites\*** In order to adapt BOP-Elites for NAS, we introduce several modifications. First, we employ truncated (one-hot) path encoding [56, 57]. In path encoding, architectures are transformed into a set of binary features indicating presence for each path of the directed acyclic graph from the input to the output. By then truncating the least-likely paths, the encoding scales linearly in the size of the cell [57] allowing for an efficient representation of architectures. Second, we substitute the Gaussian process surrogate models used in BOP-Elites with random forests [4] allowing us to model non-continuous hyperparameter spaces. Random forests have been successfully used as surrogates in BO [21, 33], often performing on a par with ensembles of neural networks [57] in the context of NAS [51, 59]. Third, we introduce a local mutation scheme similarly to the one used by the BANANAS algorithm [57] for optimizing the infill criterion EJIE: Since our aim is to find high quality solutions across all niches, we maintain an archive of the incumbent architecture in each niche and perform local mutations on each incumbent. We refer to our adjusted version as BOP-Elites\* in the remainder of the paper to emphasize the difference from the original algorithm. For the initial design, we sample architectures based on adjacency matrix encoding [56].

**Multifidelity Optimizers** For NAS, performance estimation is the computationally most expensive component [13], and almost all NAS optimizers can be made more efficient by allowing access to cheaper, lower fidelity estimates [13, 31, 14, 64]. By evaluating most architectures at lower fidelity and only promoting promising architectures to higher fidelity, many more architectures can be explored given the same total computational budget. The fidelity parameter is typically the number of epochs over which an architecture is trained.

**qdHB** One of the most prominent multifidelity optimizers is Hyperband (HB; [31]), a multi-armed bandit strategy that uses repeated Successive Halving (SH; [22]) as a subroutine to identify the best configuration (e.g., architecture) among a set of randomly sampled ones. Given an initial and maximum fidelity, a scaling parameter $\eta$, and a set of configurations of size $n$, SH evaluates all configurations on the initial smallest fidelity, then sorts the configurations by performance and only keeps the best $1/\eta$ configurations. These configurations are then trained with fidelity increased by a factor of $\eta$. This process is repeated until the maximum fidelity for a single configuration is reached. HB repeatedly runs SH with different sized sets of initial configurations called brackets. Only two inputs are required: $R$, the maximum fidelity and $\eta$, the scaling parameter that controls the proportion of configurations discarded in each round of SH. Based on these inputs, the number $s_{\max}$ and size $n_i$ of different brackets is determined. To adapt HB to the QD setting, we must track the incumbent architecture in each niche and promote configurations based on their performance within the respective niche (see Supplement B): To achieve this, we choose the top $\lfloor n_i/\eta \rfloor$ configurations

to be promoted uniformly over the $c$ niches (done in the $\text{top}_{\text{k\_qdo}}$ function), i.e., we iteratively select one of the niches uniformly at random and choose the best configuration observed so far that has yet not been selected for promotion until $\lfloor n_i/\eta \rfloor$ configurations have been selected. Note that during this procedure, it may happen that not enough configurations belonging to a specific niche have been observed yet. In this case, we choose any configuration uniformly at random over the set of all configurations that have yet to be promoted. With those modifications, we propose qdHB, as a multifidelity QD optimizer.

**BOP-ElitesHB** While HB typically shows strong anytime performance [31], it only samples configurations at random and is typically outperformed by BO methods with respect to final performance if optimizer runtime is sufficiently large [14]. BOHB [14] combines the strengths of HB and BO in a single optimizer, resulting in strong anytime performance and fast convergence. This approach employs a fidelity schedule similar to HB to determine how many configurations to evaluate at which fidelity but replaces the random selection of configurations in each HB iteration by a model-based proposal. In BOHB, a Tree Parzen Estimator [2] is used to model densities $l(A) = p(y < \alpha | A, \mathcal{D})$ and $g(A) = p(y > \alpha | A, \mathcal{D})$, and candidates are proposed that maximize the ratio $l(A)/g(A)$, which is equivalent to maximizing EI [2]. Based on BOP-Elites* and qdHB, we can now derive the QD Bayesian optimization Hyperband optimizer (BOP-ElitesHB): Instead of selecting configurations at random at the beginning of each qdHB iteration, we propose candidates that maximize the EJIE criterion. This sampling procedure is described in Supplement B.

## 4 Main Benchmark Experiments and Results

We are interested in answering the following research questions: (**RQ1**) *Does qdNAS outperform multi-objective NAS if the optimization goal is to find high-performing architectures in pre-defined niches?* (**RQ2**) *Do multifidelity qdNAS optimizers improve over full-fidelity qdNAS optimizers?* To answer these questions, we benchmark our three qdNAS optimizers – BOP-Elites*, qdHB, and BOP-ElitesHB– on the well-known NAS-Bench-101 [61] and NAS-Bench-201 [11] and compare them to three multi-objective optimizers adapted for NAS: ParEGO*, moHB*, and ParEGOHB as well as a simple Random Search[1].

**Experimental Setup** It is important to compare optimizers using analogous implementation details. We therefore use truncated path encoding and random forest surrogates throughout our experiments for all model-based optimizers. Furthermore, we use local mutations as described in [57] in order to optimize acquisition functions in BOP-Elites*, BOP-ElitesHB, ParEGO*, and ParEGOHB. To control for differences in implementation, we re-implement all optimizers and take great care in matching the original implementations.

We provide full details regarding implementation in Supplement B and only briefly introduce conceptual differences: ParEGO* is a multi-objective optimizer based on ParEGO [28] and only deviates from BOP-Elites* in that it considers a differently scalarized objective in each iteration, which is optimized using local mutations similar to the acquisition function optimization of BOP-Elites*. moHB* is an extension of HB to the multi-objective setting (promoting configurations based on non-dominated sorting with hypervolume contribution for tie breaking, for similar approaches see, e.g., [48, 49, 50, 19]). ParEGOHB is a model-based extension of moHB* that relies on the ParEGO scalarization [28] and on the same acquisition function optimization as ParEGO*.

All optimizers were evaluated on NAS-Bench-101 (Cifar-10, validation error as the first objective and the number of trainable parameters as the feature function/second objective) and NAS-Bench-201 (Cifar-10, Cifar-100, ImageNet16-120, validation error as the first objective and latency as the feature function/second objective). For multifidelity, we train architectures for 4, 12, 36, 108 epochs on NAS-Bench-101 and for 2, 7, 22, 67, 200 epochs on NAS-Bench-201 (reflecting $\eta = 3$ in the HB variants). As the optimization budget, we consider 200 full architecture evaluations (resulting in

---

[1]using adjacency matrix encoding [56]

a total budget of 21600 epochs for NAS-Bench-101 and 40000 epochs for NAS-Bench-201). For each of these four settings, we construct three different scenarios by considering different niches of interest with respect to the feature function, resulting in a total of 12 benchmark problems. In the *small/medium/large* settings, two, five and ten niches are considered, respectively. Niches are constructed to be overlapping, and boundaries are defined based on percentiles of the feature function. For the *small* setting, the boundary is given by the 50% percentile ($q_{50\%}$), effectively resulting in two niches with boundaries $[0, q_{50\%})$ and $[0, \infty)$. For the *medium* and *large* settings, percentiles indicating progressively larger niches were used, ranging from: 1% to 30% and 70% respectively. More details on the niches can be found in Supplement C. All runs were replicated 100 times.

**Results** As an anytime performance measure, we are interested in the validation error obtained for each niche, which we aggregate in a single performance measure as $\sum_{j=1}^{c} f_{\text{err}}(A_j^*)$, i.e., we consider the sum of validation errors over the best-performing architecture per niche. If a niche is empty, we assign a validation error of 100 as a penalty (this is common practice in QDO, i.e., if no solution has been found for a niche, this niche is assigned the worst possible objective function value [25]). For the final performance, we also consider the analogous test error. Figure 2 shows the anytime performance of optimizers. We observe that model-based optimizers (BOP-Elites* and ParEGO*) in general strongly outperform Random Search, and BO HB optimizers (BOP-ElitesHB and ParEGOHB) generally outperform their full-fidelity counterparts, although this effect diminishes with increasing optimization budget. In general, HB variants that do not rely on a surrogate model (qdHB and moHB*) show poor performance compared to the model-based optimizers. Moreover, especially in the *small* number of niches setting, QD strongly outperforms multi-objective optimization. Mean ranks of optimizers with respect to final validation and test performance are given in Table 1. For completeness, we also report critical differences plots of these ranks in Supplement C.

We also conducted two four-way ANOVAs on the average performance after half and all of the optimization budget is used, with the factors problem (benchmark problem), multifidelity (whether an optimizer uses multifidelity), QDO (whether an optimizer is a QD optimizer) and model-based (whether the optimizer relies on a surrogate model)[2]. For half the budget used, results indicate significant main effects of the factors multifidelity ($F(1) = 19.13, p = 0.0001$), QDO ($F(1) = 11.08, p = 0.0017$) and model-based ($F(1) = 21.13, p < 0.0001$). For all of the budget used, the significance of multifidelity diminishes, whereas the main effects of QDO ($F(1) = 18.31, p = 0.0001$) and model-based ($F(43.44), p < 0.0001$) are still significant. We can conclude that QDO in general outperforms competitors when the goal is to find high-performing architectures in pre-defined niches. Multi-fidelity optimizers improve over full-fidelity optimizers but this effect diminishes with increasing budget. Detailed results are reported in Supplement C.

Regarding efficiency, we analyzed the expected running time (ERT) of the QD optimizers given the average performance of the respective multi-objective optimizers after half of the optimization budget: For each benchmark problem, we computed the mean validation performance of each multi-objective optimizer after having spent half of its optimization budget and investigated the ERT of the analogous[3] QD optimizer. For each benchmark problem, we then computed the ratio of ERTs between multi-objective and QD optimizers and averaged them over the benchmark problems. For BOP-ElitesHB, we observe an average ERT ratio of 2.41, i.e., in expectation, BOP-ElitesHB is a factor of 2.41 faster than ParEGOHB in reaching the average performance of ParEGOHB (after half the optimization budget). For qdHB and BOP-Elites*, the average ERT ratios are 1.14 and 1.44. We conclude that all QD optimizers are more efficient than their multi-objective counterparts. More details can be found in Supplement C.

---

[2]For this analysis, we excluded qdHB and moHB* due to their lackluster performance.

[3]BOP-ElitesHB for ParEGOHB, qdHB for moHB*, and BOP-Elites* for ParEGO*

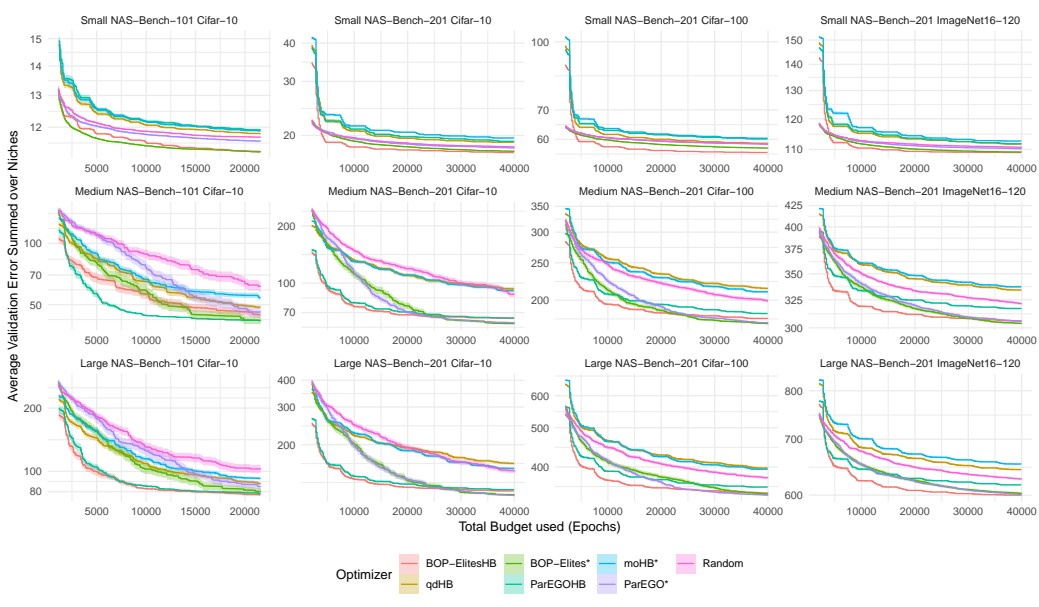

Figure 2: Anytime performance of optimizers. Ribbons represent standard errors over 100 replications. x-axis starts after 10 full-fidelity evaluations.

Table 1: Ranks of optimizers with respect to final performance, averaged over benchmark problems.

| Mean Rank (SE) | BOP-ElitesHB | qdHB | BOP-Elites* | ParEGOHB | moHB* | ParEGO* | Random |
|---|---|---|---|---|---|---|---|
| Validation | 2.08 (0.29) | 5.92 (0.26) | 1.83 (0.21) | 4.25 (0.48) | 6.42 (0.15) | 2.58 (0.31) | 4.92 (0.34) |
| Test | 1.42 (0.19) | 5.00 (0.17) | 2.08 (0.23) | 4.33 (0.50) | 6.50 (0.19) | 3.25 (0.35) | 5.42 (0.42) |

## 5 Additional Experiments and Applications

In this section, we illustrate how qdNAS can be used beyond the scenarios investigated so far and present results of additional experiments ranging from a comparison of qdNAS to multi-objective NAS on the MobileNetV3 search space to an example on how to incorporate QDO in existing frameworks such as Once-for-All [5] or how to use qdNAS for model compression.

**Benchmarks on the MobileNetV3 Search Space** We further investigated how qdNAS compares to multi-objective NAS on a search space that is frequently used in practice [20]. We consider CNNs divided into a sequence of units with feature map size gradually being reduced and channel numbers being increased. Each unit consists of a sequence of layers where only the first layer has stride 2 if the feature map size decreases and all other layers in the units have stride 1. Units can use an arbitrary number of layers (elastic depth chosen from $\{2, 3, 4\}$) and for each layer, an arbitrary number of channels (elastic width chosen from $\{3, 4, 6\}$) and kernel sizes (elastic kernel size chosen from $\{3, 5, 7\}$) can be used. Additionally, the input image size can be varied (elastic resolution ranging from 128 to 224 with a stride 4). For more details on the search space, see [5]. To allow for reasonable runtimes we use accuracy predictors (based on architectures trained and evaluated on ImageNet as described in [5]) and resource usage look-up tables of the Once-for-All module [5, 6] and construct a surrogate benchmark. As an objective function we select the validation error and as a feature function/second objective the latency (in ms) when deployed on a Samsung Note 10 (batch size of 1), or the number of FLOPS (M) used by the model. So far, we have only investigated qdNAS in the context of $k = 2$, i.e., considering one objective and one feature function. Here, we additionally consider a setting of $k = 3$, by incorporating both latency and the size of the model (in MB) as feature functions/second and third objective. We compare BOP-Elites* to ParEGO* and a

Random Search due to the accuracy predictors not supporting evaluations at multiple fidelities. We again construct three scenarios by considering different niches of interest with respect to the feature functions taking inspiration from latency and FLOPS constraints as used in [5] (details are given in Table 4 in Supplement C). Optimizers are given a total budget of 100 architecture evaluations. Figure 3 shows the anytime performance of optimizers with respect to the validation error summed over niches (averaged over 100 replications). BOP-Elites* strongly outperforms the competitors on all benchmark problems. More details are provided in Supplement D.

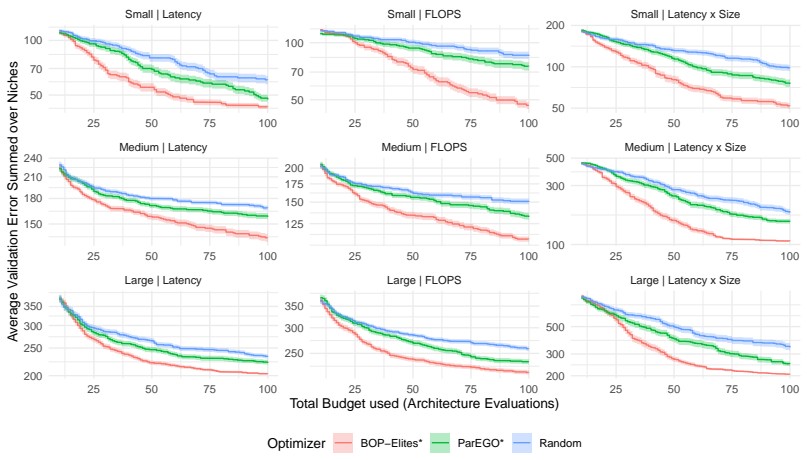

Figure 3: `MobileNetV3` search space. Anytime performance of optimizers. Ribbons represent standard errors over 100 replications. x-axis starts after 10 evaluations.

**Making Once-for-All Even More Efficient** In Once-for-All [5], an already trained supernet is searched via regularized evolution [46] for a well performing subnet that meets hardware requirements of a target device relying on an accuracy predictor and resource usage look-up tables. This is sensible if only a single solution is required, however, if various subnets meeting different constraints on the same device are desired, repeated regularized evolution is not efficient. Moreover, look-up tables do not generalize to new target devices in which case using as few as possible architecture evaluations suddenly becomes relevant again. We notice that the search for multiple architectures within Once-for-All can again be formulated as a QDO problem and therefore compare MAP-Elites [42] to regularized evolution when performing a joint search for architectures meeting different latency constraints on a Samsung Note 10. Results are given in Table 2 with MAP-Elites consistently outperforming regularized evolution, making this novel variant of Once-for-All even more efficient. More details are provided in Supplement E.

Table 2: MAP-Elites vs. regularized evolution within Once-for-All.

| Method | Best Validation Error for Different Latency Constraints | | | | | | |
|---|---|---|---|---|---|---|---|
| | $[0, 15)$ | $[0, 18)$ | $[0, 21)$ | $[0, 24)$ | $[0, 27)$ | $[0, 30)$ | $[0, 33)$ |
| Reg. Evo. | **21.57** (0.01) | 20.34 (0.02) | 19.29 (0.01) | 18.48 (0.02) | 17.81 (0.02) | 17.40 (0.02) | 17.06 (0.02) |
| MAP-Elites | 21.60 (0.01) | **20.28** (0.01) | **19.21** (0.01) | **18.39** (0.01) | **17.70** (0.01) | **17.25** (0.01) | **16.90** (0.01) |

Average over 100 replications based on the accuracy predictor of Once-for-All [5, 6]. Standard errors in parentheses. Reg. Evo. = regularized evolution.

**Applying qdNAS to Model Compression** We are interested in deploying a `MobileNetV2` across different devices that are mainly constrained by memory. For each device, we can therefore only consider models up to a fixed amount of parameters, similarly as depicted in Figure 1. Given that we have a pretrained model that achieves high performance, we want to *compress* this model exploiting redundancies in model parameters. In our application, we use the Stanford Dogs dataset [26] and

rely on the neural network intelligence (NNI; [40]) toolkit for model compression. Pruning consists of several (iterative) steps as well as re-training of the pruned architectures. Choices for the pruner itself, pruner hyperparameters, and hyperparameters controlling retraining are available and must be carefully selected to obtain optimal models (see Supplement F). We consider the number of model parameters as a proxy measure for memory requirement, yielding three overlapping niches for different devices. The pre-trained `MobileNetV2` achieves a validation error of 20.25 using around 2.34 million model parameters. We define niches with boundaries corresponding to compression rates (number of parameters after pruning) of 40% to 50%, 40% to 60%, and 40% to 70%. As the QD optimizer, we use BOP-ElitesHB and specify the number of fine-tuning epochs a pruner can use as a fidelity parameter, since fine-tuning after pruning is costly but also strongly influences final performance. After evaluating only 69 configurations (a single BOP-ElitesHB run with $\eta = 3$), we obtain high-performing pruner configurations for each niche, resulting in the performance vs. memory requirement (number of parameters) trade-offs shown in Table 3.

Table 3: Results of using BOP-ElitesHB for model compression of `MobileNetV2` on Stanford Dogs.

| Niche | Validation Error | # Params (in millions; rounded) | % (# Params$_{\text{Baseline}}$) |
|---|---|---|---|
| Niche 1 $[0.94, 1.17)$ | 31.20 | 1.13 | 48.10% |
| Niche 2 $[0.94, 1.41)$ | 29.07 | 1.29 | 54.99% |
| Niche 3 $[0.94, 1.64)$ | 27.76 | 1.62 | 68.97% |
| Baseline | 20.25 | 2.34 | 100.00% |

## 6 Conclusion

We demonstrated how multi-objective NAS can be formulated as a QDO problem that, contrary to multi-objective NAS, directly corresponds to the actual optimization problem of interest, i.e., finding high-performing architectures in different niches. We have shown how any iterative black box optimization algorithm can be adapted to the QDO setting and proposed three QDO algorithms for NAS, with two of which making use of multifidelity evaluations. In benchmark experiments, we have shown that qdNAS outperforms multi-objective NAS while simultaneously being more efficient. We furthermore illustrated how qdNAS can be used for model compression and how future NAS research can thrive on QDO. QDO is orthogonal to the NAS strategy of an algorithm and can be similarly used to extend, e.g., one-shot NAS methods.

**Limitations** The framework we describe relies on pre-defined niches, e.g., memory requirements of different devices. If niches are mis-specified or cannot be specified a priori, multi-objective NAS may outperform qdNAS. However, an initial study (see Supplement H) how qdNAS performs in a true multi-objective setting, which would correspond to unknown niches, shows little to no performance degradation depending on the choice of niches. Moreover, we only investigated the performance of qdNAS in the deterministic setting. Additionally, our multifidelity optimizers require niche membership to be unaffected by the multifidelity parameter. Finally, we mainly focused on model-based NAS algorithms that we have extended to the QDO setting.

**Broader Impact** Our work extends previous research on NAS and therefore inherits its implications on society and individuals such as potential discrimination in resulting models. Moreover, evaluating a large number of architectures is computationally costly and can introduce serious environmental issues. We have shown that qdNAS allows for finding better solutions, while simultaneously being more efficient than multi-objective NAS. As performance estimation is extremely costly in NAS, we believe that this is an important contribution towards reducing resource usage and the $CO_2$ footprint of NAS.

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

## A  Niches in NAS

In the classical QDO literature, niches are assumed to be pairwise disjoint. This implies that each architecture $A \in \mathcal{A}$ yields feature function values $f_i(A), i \geq 2$ that map to a single niche (or none). In practice, this does not necessarily have to be the case though, as an architecture can belong to multiple niches. For example, when considering memory or latency constraints, a model with lower latency or lower memory requirements can always be used in settings that allow for accommodating slower or larger models. This is illustrated in Figure 4. Note that we index niches in the disjoint scenario in Figure 4 with two indices, to highlight that some niches share the same boundaries on a given feature function (e.g., $N_{1,1}$ and $N_{2,1}$ share the same latency boundaries and only differ with respect to the memory boundaries). In this paper, we mainly investigated the scenario of nested niches. The setting for QDO in the NAS context as described in Figure 1 in the main paper is given by the search of models for deployment on multiple different end-user devices. Similarly, qdNAS can also be applied in the context of searching for models for deployment on a single end-user device, meeting different constraints, e.g., as illustrated in Section 5 (Benchmarks on the `MobileNetV3` Search Space) in the main paper. Typically, relevant boundaries of feature functions that form niches naturally arise given the target device(s) and concrete application at hand.

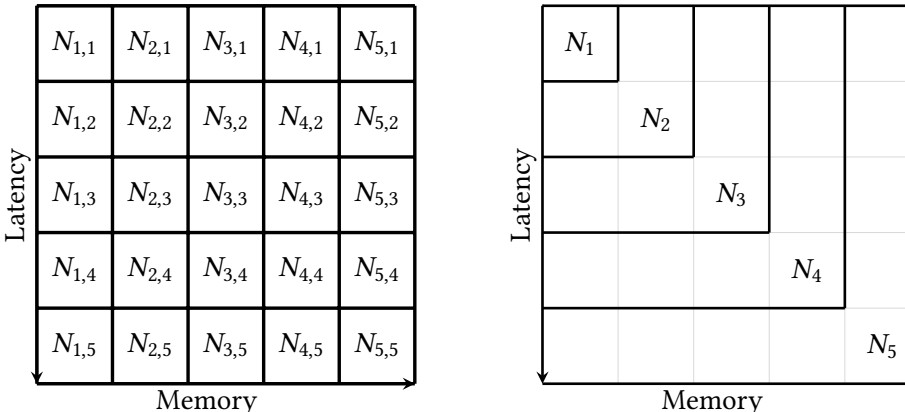

Figure 4: Disjoint (left) and nested (right) niches.

## B  Optimizers

In this section, we provide additional information on optimizers used throughout this paper. Algorithm 2 illustrates a generic iterative single-objective optimizer in pseudo code.

**qdHB** Algorithm 3 presents qdHB in pseudo code. qdHB requires only $R$ (maximum fidelity) and $\eta$ (scaling parameter) as input parameters and proceeds to determine the maximum number of brackets $s_{\max}$ and the approximate total resources $B$ which each bracket is assigned. In each bracket $s$, the number of configurations $n$ and the fidelity $r$ at which they should be evaluated is calculated and these parameters are used within the SH subroutine. The central step within the SH subroutine is the selection of the $\lfloor n_i/\eta \rfloor$ configurations that should be promoted to the next stage. Here, the $\text{top}_{\text{k\_qdo}}$ function (highlighted in grey) works as follows: We iteratively select one of the niches uniformly at random and choose the best configuration within this niche observed so far that has yet not been selected for promotion. This procedure is repeated until $\lfloor n_i/\eta \rfloor$ configurations have been selected in total. If not enough configurations belonging to a specific niche have been observed so far, we choose any configuration uniformly at random over the set of all configurations that have yet to be promoted. Note that feature functions and thereupon derived niche membership are assumed to be unaffected by the multifidelity parameter. Niche membership is determined by the

---

**Algorithm 2:** Generic pseudo code for an iterative single-objective optimizer.

---

**Input** : $f_1, \mathcal{D}_{\text{design}}, n_{\text{total}}$
**Result**: $A^*$

1  $\mathcal{D} \leftarrow \mathcal{D}_{\text{design}}$
2  $A^* \leftarrow \arg\min_{A \in \mathcal{D}} f_1(A)$ # initial incumbent based on archive
3  **for** $n \leftarrow 1$ **to** $n_{\text{total}}$ **do**
4       Propose a new candidate $A^\star$
5       Evaluate $y \leftarrow f_1(A^\star)$
6       **if** $y < f_1(A^*)$ **then**
7          $A^* \leftarrow A^\star$ # update incumbent
8       **end**
9       $\mathcal{D} \leftarrow \mathcal{D} \cup \{(A^\star, y)\}$
10 **end**

---

`get_niche_membership` function which simply checks for each niche whether feature values of an architecture are within the respective niche boundaries. Moreover, we assume that all evaluations are written into an archive similarly as in Algorithm 1 in the main paper which allows us to return the best configuration per niche as the final result. Note that in practice, evaluating all stages of brackets with the same budget instead of iterating over brackets (like in the original HB implementation) can be more efficient. We use this scheduling variant throughout our benchmark experiments and application study. More details regarding our implementation can be obtained via https://github.com/slds-lmu/qdo_nas.

---

**Algorithm 3:** Quality Diversity Hyperband (qdHB).

---

**Input** : $R, \eta$ # maximum fidelity and scaling parameter
**Result**: Best configuration per niche

1  $s_{\text{max}} = \lfloor \log_\eta(R) \rfloor, B = (s_{\text{max}} + 1)R$
2  **for** $s \in \{s_{\text{max}}, s_{\text{max}} - 1, \ldots, 0\}$ **do**
3       $n = \lceil \frac{B}{R} \frac{\eta^s}{(s+1)} \rceil, r = R\eta^{-s}$
4       # begin SH with $(n, r)$ inner loop
5       $\mathbf{A} = \texttt{sample\_configuration}(n)$
6       $\mathbf{Z} = \{(f_i(A), \ldots, f_k(A)) : A \in \mathbf{A}, i \in \{2, \ldots, k\}\}$ # evaluate feature functions
7       $\mathbf{N} = \texttt{get\_niche\_membership}(\mathbf{A}, \mathbf{Z})$
8       **for** $i \in \{0, \ldots, s\}$ **do**
9          $n_i = \lfloor n\eta^{-i} \rfloor$
10         $r_i = r\eta^i$
11         $\mathbf{Y} = \{f_1(A, r_i) : A \in \mathbf{A}\}$ # evaluate objective function
12         $\mathbf{A} = \texttt{top}_{\texttt{k\_qdo}}(\mathbf{A}, \mathbf{Y}, \mathbf{N}, \lfloor n_i/\eta \rfloor)$
13      **end**
14 **end**

---

**BOP-ElitesHB** In Algorithm 4 we describe the sampling procedure (for a single configuration) used in BOP-ElitesHB in pseudo code. In contrast to the original BOHB algorithm, we use random forest as surrogate models, similarly as done in SMAC-HB [33]. Throughout our benchmark experiments and application study we set $\rho = 0$. Furthermore, we employ a variant that directly proposes batches of size $n$. This can be done by simply sorting all candidate architectures obtained via local mutation of the incumbent architectures of each niche within the acquisition function

optimization step by their EJIE values and selecting the top $n$ candidate architectures. Note that surrogate models are fitted on all available data contained in the current archive (this includes the multifidelity parameter) and predictions are obtained with respect to the fidelity parameter set to the current fidelity level. More details regarding our implementation can be obtained via `https://github.com/slds-lmu/qdo_nas`.

---

**Algorithm 4**: Sampling procedure in BOP-ElitesHB.

---

**Input** : $\rho$ # fraction of configurations sampled at random
**Result**: Next configuration to evaluate

1 **if** rand() $< \rho$ **then**
2      **return** sample_configuration(1)
3 **else**
4      $A^\star \leftarrow \arg\max_{A \in \mathcal{A}} \alpha_{\text{EJIE}}(A)$ # Equation (1)
5      **return** $A^\star$
6 **end**

---

**ParEGO\*** ParEGO [28] is a multi-objective model-based optimizer that at each iteration scalarizes the objective functions differently using the augmented Tchebycheff function. First, the $k$ objectives are normalized and at each iteration a weight vector $\lambda$ is drawn uniformly at random from the following set of $\binom{s+k-1}{k-1}$ different weight vectors[4]:

$$\left\{ \lambda = (\lambda_1, \lambda_2, \ldots, \lambda_k) \mid \sum_{i=1}^{k} \lambda_i = 1 \ \wedge \ \lambda_i = \frac{l}{s}, l \in \{0, \ldots, s\} \right\}.$$

The scalarization is then obtained via $f_\lambda(A) = \max_{i=1}^{k} (\lambda_i \cdot f_i(A)) + \gamma \sum_{i=1}^{k} \lambda_i \cdot f_i(A)$, where $\gamma$ is a small positive value (in our benchmark experiments we use 0.05). In ParEGO\* we use the same truncated path encoding as in BOP-Elites\* as well as a random forest surrogate modeling the scalarized objective function. For optimizing the EI, we use a local mutation scheme similarly to the one utilized by BANANAS [57], adapted for the multi-objective setting (conceptually similar to the one proposed by [19]): For each Pareto optimal architecture in the current archive, we obtain candidate architectures via local mutation and out of all these candidates we select the architecture with the largest EI for evaluation. More details regarding our implementation can be obtained via `https://github.com/slds-lmu/qdo_nas`.

**moHB\*** moHB\* [28] is an extension of HB to the multi-objective setting. The optimizer follows the basic HB routine except for the selection mechanism of configurations that should be promoted to the next stage: Configurations are promoted based on non-dominated sorting with hypervolume contribution for tie breaking. For similar approaches, see [48, 49, 50, 19]. In our benchmark experiments we again use a scheduling variant that evaluates all stages of brackets with the same budget instead of iterating over brackets. More details regarding our implementation can be obtained via `https://github.com/slds-lmu/qdo_nas`.

**ParEGOHB** ParEGOHB combines BO with moHB\* by using the same scalarization as ParEGO\*. Instead of selecting configurations at random at the beginning of each moHB\* iteration, ParEGOHB proposes candidates that maximize the EI with respect to the scalarized objective. In our benchmark experiments we again set $\rho = 0$ (fraction of configurations sampled uniformly at random) and employ a variant that directly proposes batches of size $n$. Note that surrogate models are fitted on all available data contained in the current archive (this includes the multifidelity parameter) and predictions are obtained with respect to the fidelity parameter set to the current fidelity level. More details regarding our implementation can be obtained via `https://github.com/slds-lmu/qdo_nas`.

---

[4]note that $s$ simply determines the number of different weight vectors

Table 4: Niches and their boundaries used throughout all benchmark experiments.

| Benchmark | Dataset | Niches | Niche Boundaries | | | | | | | | | |
|---|---|---|---|---|---|---|---|---|---|---|---|---|
| | | | Niche 1 | Niche 2 | Niche 3 | Niche 4 | Niche 5 | Niche 6 | Niche 7 | Niche 8 | Niche 9 | Niche 10 |
| NAS-Bench-101 # Params | Cifar-10 | Small | $[0, 5356682)$ | $[0, \infty)$ | - | - | - | - | - | - | - | - |
| | | Medium | $[0, 650520)$ | $[0, 1227914)$ | $[0, 1664778)$ | $[0, 3468426)$ | $[0, \infty)$ | - | - | - | - | - |
| | | Large | $[0, 650520)$ | $[0, 824848)$ | $[0, 1227914)$ | $[0, 1664778)$ | $[0, 2538506)$ | $[0, 3468426)$ | $[0, 3989898)$ | $[0, 5356682)$ | $[0, 8118666)$ | $[0, \infty)$ |
| NAS-Bench-201 Latency | Cifar-10 | Small | $[0, 0.015000444871408)$ | $[0, \infty)$ | - | - | - | - | - | - | - | - |
| | | Medium | $[0, 0.00856115)$ | $[0, 0.01030767)$ | $[0, 0.01143533)$ | $[0, 0.01363741)$ | $[0, \infty)$ | - | - | - | - | - |
| | | Large | $[0, 0.00856115)$ | $[0, 0.00893427)$ | $[0, 0.01030767)$ | $[0, 0.01143533)$ | $[0, 0.01250159)$ | $[0, 0.01363741)$ | $[0, 0.01429903)$ | $[0, 0.01500044)$ | $[0, 0.01660615)$ | $[0, \infty)$ |
| | Cifar-100 | Small | $[0, 0.0159673188862048)$ | $[0, \infty)$ | - | - | - | - | - | - | - | - |
| | | Medium | $[0, 0.00919228)$ | $[0, 0.01138714)$ | $[0, 0.01232998)$ | $[0, 0.01475572)$ | $[0, \infty)$ | - | - | - | - | - |
| | | Large | $[0, 0.00919228)$ | $[0, 0.00957457)$ | $[0, 0.01138714)$ | $[0, 0.01232998)$ | $[0, 0.01327515)$ | $[0, 0.01475572)$ | $[0, 0.01534633)$ | $[0, 0.01596732)$ | $[0, 0.01768237)$ | $[0, \infty)$ |
| | ImageNet16-120 | Small | $[0, 0.014301609992981)$ | $[0, \infty)$ | - | - | - | - | - | - | - | - |
| | | Medium | $[0, 0.00767465)$ | $[0, 0.0094483)$ | $[0, 0.01054566)$ | $[0, 0.01271056)$ | $[0, \infty)$ | - | - | - | - | - |
| | | Large | $[0, 0.00767465)$ | $[0, 0.00826192)$ | $[0, 0.0094483)$ | $[0, 0.01054566)$ | $[0, 0.01173623)$ | $[0, 0.01271056)$ | $[0, 0.01352221)$ | $[0, 0.01430161)$ | $[0, 0.01595311)$ | $[0, \infty)$ |
| MobileNetV3 Latency | ImageNet | Small | $[0, 17.5)$ | $[0, 30)$ | - | - | - | - | - | - | - | - |
| | | Medium | $[0, 15)$ | $[0, 20)$ | $[0, 25)$ | $[0, 30)$ | $[0, 35)$ | - | - | - | - | - |
| | | Large | $[0, 17)$ | $[0, 19)$ | $[0, 21)$ | $[0, 23)$ | $[0, 25)$ | $[0, 27)$ | $[0, 29)$ | $[0, 31)$ | $[0, 33)$ | $[0, 35)$ |
| MobileNetV3 FLOPS | ImageNet | Small | $[0, 150)$ | $[0, 400)$ | - | - | - | - | - | - | - | - |
| | | Medium | $[0, 150)$ | $[0, 200)$ | $[0, 250)$ | $[0, 300)$ | $[0, 400)$ | - | - | - | - | - |
| | | Large | $[0, 150)$ | $[0, 175)$ | $[0, 200)$ | $[0, 225)$ | $[0, 250)$ | $[0, 275)$ | $[0, 300)$ | $[0, 325)$ | $[0, 350)$ | $[0, 400)$ |
| MobileNetV3 Latency × Size | ImageNet | Small | $[0, 20) \times [0, 20)$ | $[0, 35) \times [0, 20)$ | - | - | - | - | - | - | - | - |
| | | Medium | $[0, 20) \times [0, 20)$ | $[0, 25) \times [0, 20)$ | $[0, 30) \times [0, 20)$ | $[0, 35) \times [0, 20)$ | $[0, 40) \times [0, 20)$ | - | - | - | - | - |
| | | Large | $[0, 20) \times [0, 20)$ | $[0, 23) \times [0, 20)$ | $[0, 26) \times [0, 20)$ | $[0, 29) \times [0, 20)$ | $[0, 32) \times [0, 20)$ | $[0, 35) \times [0, 20)$ | $[0, 38) \times [0, 20)$ | $[0, 41) \times [0, 20)$ | $[0, 44) \times [0, 20)$ | $[0, 47) \times [0, 20)$ |

## C Additional Benchmark Details and Results

In this section, we provide additional details and analyses with respect to our main benchmark experiments. Table 4 summarizes all niches and their boundaries used throughout our benchmarks (including the additional ones on the `MobileNetV3` search space).

The following results extends the results reported for the main benchmark experiments. Critical differences plots ($\alpha = 0.05$) of optimizer ranks (with respect to final performance) are given in Figure 5. Friedman tests ($\alpha = 0.05$) that were conducted beforehand indicated significant differences in ranks for both the validation ($\chi^2(6) = 53.46, p < 0.001$) and test performance ($\chi^2(6) = 52.14, p < 0.001$). However, note that critical difference plots based on the Nemenyi test are underpowered if only few optimizers are compared on few benchmark problems.

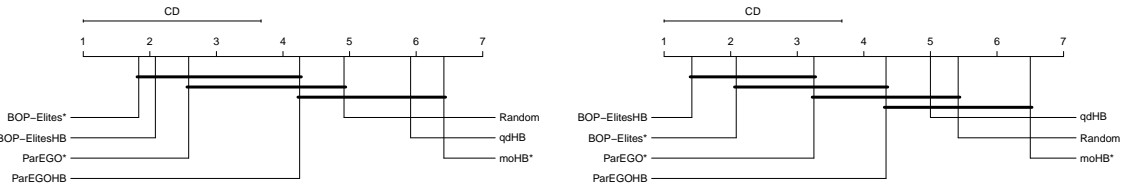

(a) Validation error summed over niches.  (b) Test error summed over niches.

Figure 5: Critical differences plots of the ranks of optimizers.

Figure 6 and Figure 7 show the average best validation and test performance for each niche for each optimizer on each benchmark problem.

Table 5 summarizes results of a four way ANOVA on the average performance (validation error summed over niches) of BOP-ElitesHB, BOP-Elites*, ParEGOHB, ParEGO* and Random after having used half of the total optimization budget. Prior to conducting the ANOVA, we checked the ANOVA assumptions (normal distribution of residuals and homogeneity of variances) and found no violation of assumptions. The factors are given as follows: Problem indicates the benchmark problem (e.g., NAS-Bench-101 on Cifar-10 with small number of niches), multifidelity denotes if the optimizer uses multifidelity (`TRUE` for BOP-ElitesHB and ParEGOHB), QDO denotes whether the optimizer is a QD optimizer (`TRUE` for BOP-ElitesHB and BOP-Elites*) and model-based denotes whether the optimizer relies on a surrogate model (`TRUE` for BOP-ElitesHB, BOP-Elites*, ParEGOHB and ParEGO*). All main effects are significant at an $\alpha$ level of 0.05. We also computed confidence intervals based on Tukey's Honest Significant Difference method for the estimated differences between factor levels: Multifidelity $-12.45[-18.19 - 6.72]$, QDO $-9.34[-15.08, -3.61]$, model-based $-13.55[-20.57, -6.52]$. Note that the negative sign indicates a decrease in the average validation error summed over niches.

Table 5: Results of a four way ANOVA on the average performance (validation error summed over niches) after having used half of the total optimization budget. Type II sums of squares.

|  | Df | Sum Sq | Mean Sq | F value | Pr(>F) |
|---|---|---|---|---|---|
| Problem | 11 | 1810569.79 | 164597.25 | 1410.16 | 0.0000 |
| Multifidelity | 1 | 2233.44 | 2233.44 | 19.13 | 0.0001 |
| QDO | 1 | 1293.41 | 1293.41 | 11.08 | 0.0017 |
| Model-Based | 1 | 2466.45 | 2466.45 | 21.13 | 0.0000 |
| Residuals | 45 | 5252.51 | 116.72 |  |  |

We conducted a similar ANOVA on the final performance of optimizers (Table 6). Prior to conducting the ANOVA, we checked the ANOVA assumptions (normal distribution of residuals and homogeneity of variances) and found no violation of assumptions. While the effects of QDO

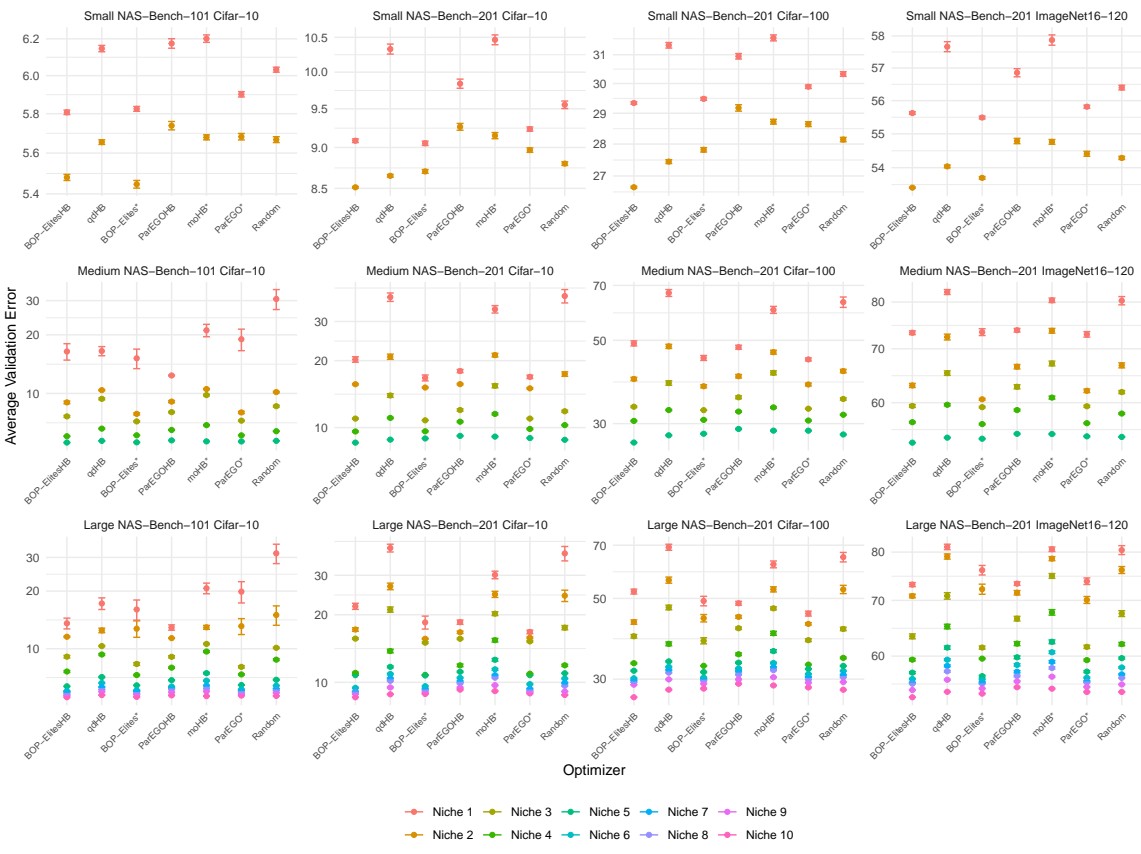

Figure 6: Best solution found in each niche with respect to validation performance. Bars represent standard errors over 100 replications.

and model-based are still significant at an $\alpha$ level of 0.05, the effect of multifidelity no longer is, indicating that full-fidelity optimizer caught up in performance (which is the expected behavior). We again computed confidence intervals based on Tukey's Honest Significant Difference method for the estimated differences between factor levels: QDO $-6.85[-10.12 - 3.58]$, model-based $-11.08[-15.08, -7.07]$.

Table 6: Results of a four way ANOVA on the average final performance (validation error summed over niches). Type II sums of squares.

|  | Df | Sum Sq | Mean Sq | F value | Pr(>F) |
|---|---|---|---|---|---|
| Problem | 11 | 1724557.85 | 156777.99 | 4130.33 | 0.0000 |
| Multifidelity | 1 | 97.76 | 97.76 | 2.58 | 0.1155 |
| QDO | 1 | 695.13 | 695.13 | 18.31 | 0.0001 |
| Model-Based | 1 | 1648.94 | 1648.94 | 43.44 | 0.0000 |
| Residuals | 45 | 1708.10 | 37.96 |  |  |

We analyzed the ERT of the QD optimizers given the average performance of the respective multi-objective optimizers after half of the optimization budget. For each benchmark problem, we computed the mean validation performance of each multi-objective optimizer after having spent half of its optimization budget and investigated the analogous QD optimizer. We then computed the ratio of ERTs between multi-objective and QD optimizers (see Table 7).

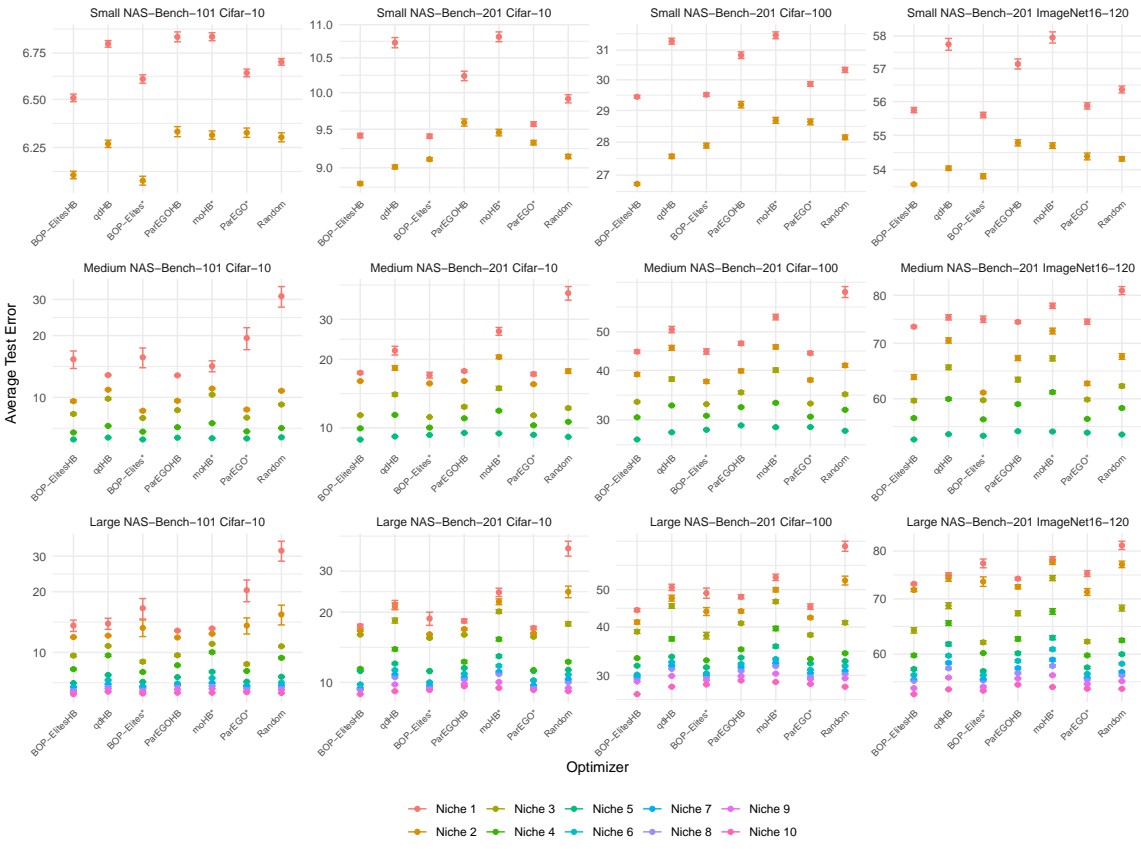

Figure 7: Best solution found in each niche with respect to test performance. Bars represent standard errors over 100 replications.

## D Details on Benchmarks on the `MobileNetV3` Search Space

In this section, we provide additional details regarding our benchmarks on the MobileNetV3 Search Space. We use `ofa_mbv3_d234_e346_k357_w1.2` as a pretrained supernet and rely on accuracy predictors and latency/FLOPS look-up tables as provided by [6]. The search space of architectures is the same as used in [5]. For the model-based optimizers we employ the following encoding of architectures: Given an architecture, we encode each layer in the neural network into a one-hot vector based on its kernel size and expand ratio and we assign zero vectors to layers that are skipped. Besides, we have an additional one-hot vector that represents the input image size. We concatenate these vectors into a large vector that represents the whole neural network architecture and input image size. This is the same encoding as used by [5]. Acquisition function optimization is performed by sampling 1000 architectures uniformly at random.

## E Details on Making Once-for-All Even More Efficient

In this section, we provide additional details regarding replacing regularized evolution with MAP-Elites within Once-for-All. We use `ofa_mbv3_d234_e346_k357_w1.2` as a pretrained supernet and rely on accuracy predictors and latency look-up tables as provided by [6]. Seven niches were defined via the following latency constraints (in ms): $[0, 15], [0, 18], [0, 21], [0, 24], [0, 27], [0, 30], [0, 33]$. Regularized evolution is run with an initial population of size 100 for 71 generations [5] resulting in

---
[5]this is exactly $\lceil (50100 - 7 \cdot 100)/(7 * 100) \rceil$ with 50100 being the budget MAP-Elites is allowed to use

Table 7: ERT ratios of multi-objective and QD optimizers to reach the average performance (after half of the optimization budget) of the respective multi-objective optimizer.

| Benchmark | Dataset | Niches | ERT Ratio | | |
|---|---|---|---|---|---|
| | | | ParEGOHB/ BOP-ElitesHB | moHB*/ qdHB | ParEGO*/ BOP-Elites* |
| NAS-Bench-101 | Cifar-10 | Small | 3.94 | 1.19 | 1.99 |
| | | Medium | 0.76 | 1.43 | 1.85 |
| | | Large | 1.47 | 1.20 | 1.58 |
| NAS-Bench-201 | Cifar-10 | Small | 4.31 | 1.96 | 1.45 |
| | | Medium | 0.94 | 0.73 | 1.34 |
| | | Large | 1.04 | 0.72 | 1.35 |
| | Cifar-100 | Small | 4.82 | 1.77 | 1.46 |
| | | Medium | 1.35 | 0.67 | 1.42 |
| | | Large | 1.57 | 0.78 | 0.98 |
| | ImageNet16-120 | Small | 4.30 | 1.20 | 1.73 |
| | | Medium | 2.31 | 0.93 | 1.14 |
| | | Large | 2.11 | 1.15 | 0.96 |

7200 architecture evaluations per latency constraint and 50400 architecture evaluations in total. We use a mutation probability of 0.1, a mutation ratio of 0.5 and a parent ratio of 0.25. MAP-Elites searches for optimal architectures jointly for the seven niches and is configured to use a population of size 100 and is run for 500 generations, resulting in 50100 architecture evaluations in total. The number of generations for each regularized evolution run and the MAP-Elites run were chosen in a way so that the total number of architecture evaluations is roughly the same for both methods. We again use a mutation probability of 0.1. Note that the basic MAP-Elites (as used by us) does not employ any kind of crossover. We visualize the best validation error obtained for each niche in Figure 8 (left). MAP-Elites outperforms regularized evolution in almost every niche, making this variant of Once-for-All even more efficient. In the scenario of using Once-for-All for new devices, look-up tables do not generalize and the need for using as few as possible architecture evaluations is of central importance. To illustrate how MAP-Elites compares to regularized evolution in this scenario, we reran the experiments above but this time we used a population of size 50 and 100 generations for MAP-Elites (and therefore 14 generations for each run of regularized evolution). Results are illustrated in Figure 8 (right). Again, MAP-Elites generally outperforms regularized evolution.

## F  Details on Applying qdNAS to Model Compression

In this section, we provide additional details regarding our application of qdNAS to model compression. BOP-ElitesHB was slightly modified due to the natural tabular representation of the search space. Instead of using a truncated path encoding we simply use the tabular representation of parameters. To optimize the EJIE during the acquisition function optimization step we employ a simple Random Search, sampling 10000 configurations uniformly at random and proposing the configuration with the largest EJIE. Table 8 shows the search space used for tuning NNI pruners on `MobileNetV2`.

## G  Analyzing the Effect of the Choice of the Surrogate Model and Acquisition Function Optimizer

In this section, we present results of a small ablation study regarding the effect of the choice of the surrogate model and acquisition function optimizer. In the main benchmark experiments, we observed that our qdNAS optimizers sometimes fail to find any architecture belonging to a certain

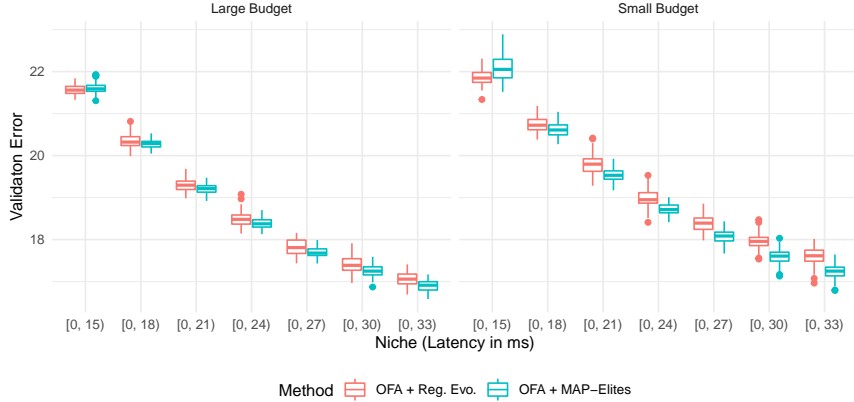

Figure 8: Regularized evolution vs. MAP-Elites within Once-for-All. Left: Large budget of total architecture evaluations. Right: Small budget of total architecture evaluations. Boxplots are based on 100 replications.

Table 8: Search space for NNI pruners on `MobileNetV2`.

| Hyperparameter | Type | Range | Info |
|---|---|---|---|
| pruning_mode | categorical | {conv0, conv1, conv2, conv1andconv2, all} | |
| pruner_name | categorical | {l1, l2, slim, agp, fpgm, mean_activation, apoz, taylorfo} | |
| sparsity | continuous | [0.4, 0.7] | |
| agp_pruning_alg | categorical | {l1, l2, slim, fpgm, mean_activation, apoz, taylorfo} | |
| agp_n_iters | integer | [1, 100] | |
| agp_n_epochs_per_iter | integer | [1, 10] | |
| slim_sparsifying_epochs | integer | [1, 30] | |
| speed_up | boolean | {TRUE, FALSE} | |
| finetune_epochs | integer | [1, 27] | fidelity |
| learning_rate | continuous | [1e-06, 0.01] | log |
| weight_decay | continuous | [0, 0.1] | |
| kd | boolean | {TRUE, FALSE} | |
| alpha | continuous | [0, 1] | |
| temp | continuous | [0, 100] | |

"agp_pruning_alg", "agp_n_iters", and "agp_n_epochs_per_iter" depend on "pruner_name" being "agp". "slim_sparsifying_epochs" depends on "pruner_name" being "slim". "alpha" and "temp" depend on "kd" being "TRUE". "log" in the Info column indicates that this parameter is optimized on a logarithmic scale.

niche (even after having used all available budget). This was predominantly the case for the very small niches in the medium and large number of niches settings (i.e., Niche 1, 2 or 3). Figure 9 shows the relative frequency of niches missed by optimizers (over 100 replications). Note that for the small number of niches settings, relative frequencies are all zero and therefore omitted. In general, model-based multifidelity variants perform better than the full-fidelity optimizers and QD optimizers sometimes perform worse than multi-objective optimizers.

We hypothesized that this could be caused by the choice of the surrogate model used for the feature functions: A random forest cannot properly extrapolate values outside the training set and therefore, if the initial design does not contain an architecture for a certain niche, the optimizer may fail to explore relevant regions in the feature space. We therefore conducted a small ablation study on the NAS-Bench-101 Cifar-10 medium number of niches benchmark problem. BOP-Elites* was configured to either use a random forest (as before) or an ensemble of feed-forward neural networks[6] (as used by BANANAS [57]) as a surrogate model for the feature function. Moreover, we

---

[6] with an ensemble size of five networks

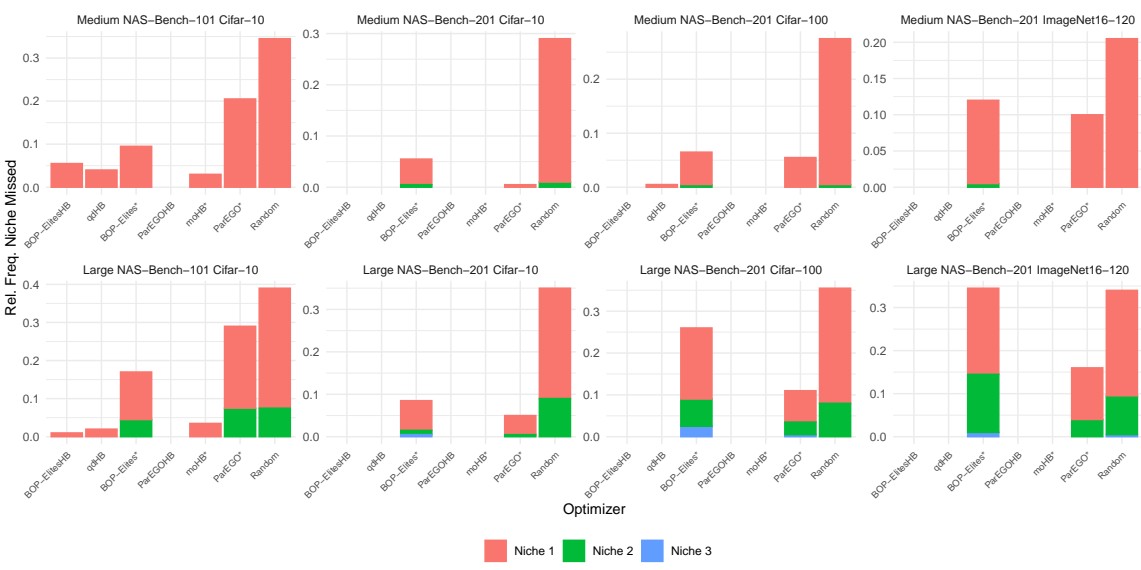

Figure 9: Relative frequency of niches missed by optimizers over 100 replications. For the small number of niches settings, relative frequencies are all zero and therefore omitted.

varied the acquisition function optimizer between a local mutation (as before) or a simple Random Search (generating the same number of candidate architectures but sampling them uniformly at random using adjacency matrix encoding). Optimizers were given a budget of 100 full architecture evaluations and runs were replicated 30 times. Figure 10 shows the anytime performance of these BOP-Elites* variants. We observe that switching to an ensemble of neural networks as a surrogate model for the feature function results in a performance boost which can be explained by the fact that this BOP-Elites* variant no longer struggles with finding solutions in the smallest niche. The relative frequencies of a solution for Niche 1 being missing are: 26.67% for the random forest + Random Search, 16.67% for the random forest + mutation, 3.33% for the ensemble of neural networks + Random Search, and 3.33% for the ensemble of neural networks + mutation. Regarding the other niches, a solution is always found. Results also suggest that the choice of the acquisition function optimizer may be more important in case of using a random forest as a surrogate model for the feature function.

## H  Judging Quality Diversity Solutions by Means of Multi-Objective Performance Indicators

In this section, we analyze the performance of our qdNAS optimizers in the context of a multi-objective optimization setting. As an example, suppose that niches were mis-specified and the actual solutions (best architecture found for each niche) returned by the QD optimizers are no longer of interest. We still could ask the question of how well QDO performs in solving the multi-objective optimization problem. To answer this question, we evaluate the final performance of all optimizers compared in Section 4 by using multi-objective performance indicators. Figure 11 shows the average Hypervolume Indicator (the difference in hypervolume between the resulting Pareto front approximation of an optimizer for a given run and the best Pareto front approximation found over all optimizers and replications). For these computations, the feature function was transformed to the logarithmic scale for the NAS-Bench-101 problems. As nadir points we used $(100, \log(49979275))'$ for the NAS-Bench-101 problems and $(100, 0.0283)'$ for the NAS-Bench-201 problems obtained by taking the theoretical worst validation error of 100 and feature function upper limits as found in the tabular benchmarks (plus some additional small numerical tolerance).

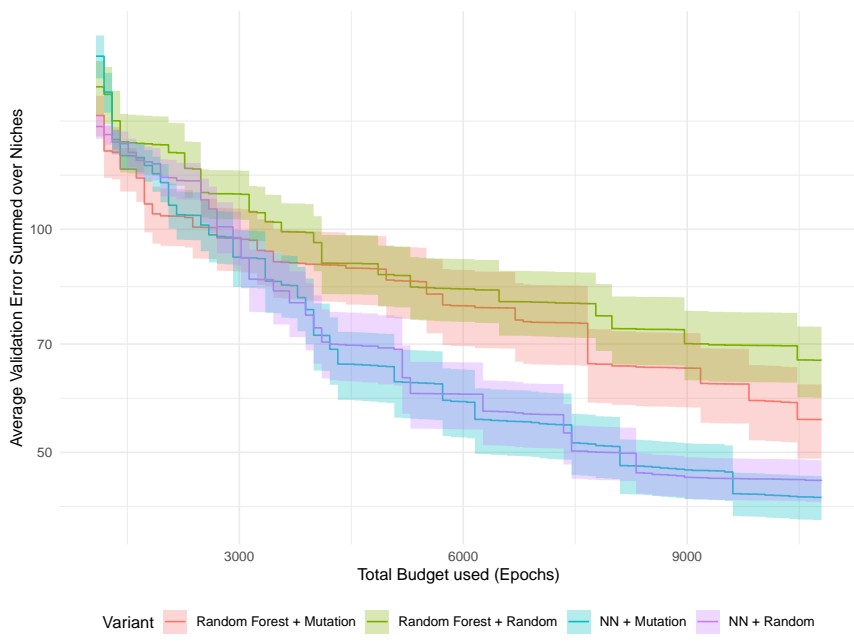

Figure 10: Anytime performance of BOP-Elites* variants configured to either use a random forest or an ensemble of neural networks as a surrogate model for the feature function crossed with either using a local mutation or a Random Search as acquisition function optimizer. NAS-Bench-101 Cifar-10 medium number of niches benchmark problem. Ribbons represent standard errors over 30 replications. x-axis starts after 10 full-fidelity evaluations.

Note that for all optimizers which are not QD optimizers, results with respect to the different number of niches settings (small vs. medium vs. large) are only statistical replications because these optimizers are not aware of the niches. We observe that ParEGOHB and ParEGO* perform well but BOP-ElitesHB also shows good performance in the medium and large number of niches settings. This is the expected behavior, as the number and nature of the niches directly corresponds to the ability of qdNAS optimizers to search along the whole Pareto front, i.e., in the small number of niches settings, qdNAS optimizers have no intention to explore.

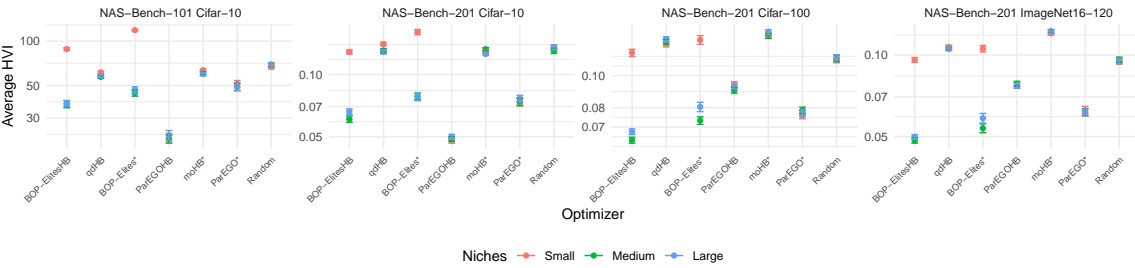

Figure 11: Average Hypervolume Indicator. Bars represent standard errors over 100 replications.

Critical differences plots ($\alpha = 0.05$) of optimizer ranks (with respect to the Hypervolume Indicator) are given in Figure 12. A Friedman test ($\alpha = 0.05$) that was conducted beforehand indicated significant differences in ranks ($\chi^2(6) = 41.61, p < 0.001$). Again, note that critical difference plots based on the Nemenyi test are underpowered if only few optimizers are compared on few benchmark problems.

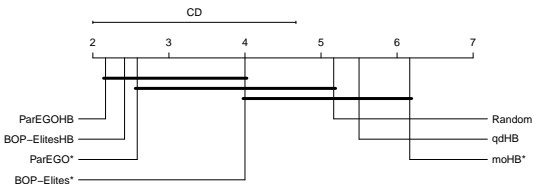

Figure 12: Critical differences plot of the ranks of optimizers with respect to the Hypervolume Indicator.

In Figure 13 we plot the average Pareto front (over 100 replications) for BOP-Elites*, ParEGO* and Random. The average Pareto fronts of BOP-Elites* and ParEGO* are relatively similar, except for the small number of niches settings, where ParEGO* has a clear advantage. Summarizing, qdNAS optimizers can also perform well in a multi-objective optimization setting, but their performance strongly depends on the number and nature of niches.

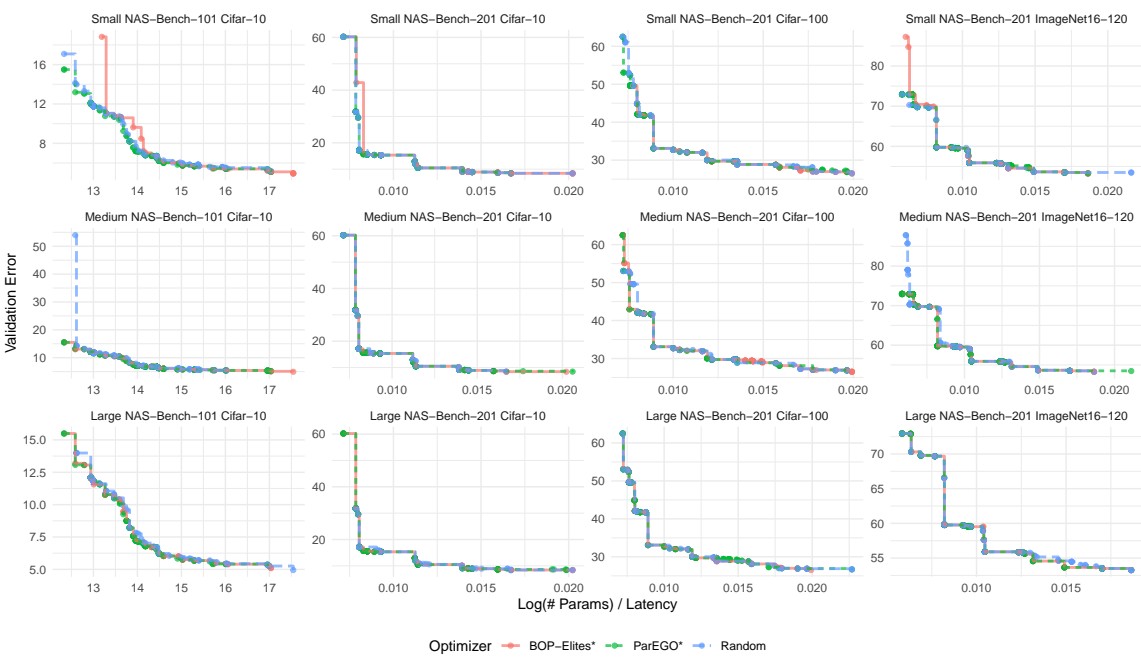

Figure 13: Average Pareto front (over 100 replications) for BOP-Elites*, ParEGO* and Random.

## I  Technical Details

Benchmark experiments were run on NAS-Bench-101 (Apache-2.0 License) [61] and NAS-Bench-201 (MIT License) [11]. More precisely, we used the `nasbench_full.tfrecord` data for NAS-Bench-101 and the `NAS-Bench-201-v1_1-096897.pth` data for NAS-Bench-201. Parts of our code rely on code released in Naszilla (Apache-2.0 License) [56, 57, 58]. For our benchmarks on the `MobileNetV3` search space we used the Once-for-All module [6] released under the MIT License. We rely on `ofa_mbv3_d234_e346_k357_w1.2` as a pretrained supernet and accuracy predictors and resource usage look-up tables as provided by [6]. NNI is released under the MIT License [40]. Stanford Dogs is released under the MIT License [26]. Figure 1 in the main paper has been designed using resources from `Flaticon.com`. Benchmark experiments were run on Intel Xeon E5-2697 instances taking around 939 CPU hours (benchmarks and ablation studies). The model compression application was performed on an NVIDIA DGX A100 instance taking around 3 GPU

days. Total emissions are estimated to be an equivalent of 72.30 kg $CO_2$. All our code is available at https://github.com/slds-lmu/qdo_nas.

