# OpenReview forum: "Tackling Neural Architecture Search With Quality Diversity Optimization"
_automl.cc/AutoML/2022/Track/Main — AutoML-Conf 2022 (Main Track)_

### Official Review · Reviewer_agqQ · 2022-04-04

**Potential Impact On The Field Of Automl Rating:** 4
**Technical Quality And Correctness Rating:** 3
**Clarity Rating:** 4

**Summary Of Contributions:**

- The paper discusses neural architecture search (NAS) as a quality diversity optimization problem.
- Instead of treating it as a multi-objective optimization problem, a primary objective function (in the context of NAS usually the model performance, e.g., accuracy) is optimized s.t. it is optimal within one niche, e.g., such that a certain memory requirement is fulfilled.
- The authors present a new acquisition function for Bayesian Optimization, the expected joint improvement of elites, which extends expected improvement to the given problem setting.
- The authors present three algorithms, BOP-Elites*, qdHB, and BOP-ElitesHB, and show their performance on several NAS-Bench benchmarks.

**Clarity:**

Overall, the paper is clear and well motivated.

The description of the truncated path encoding is rather short. I would assume that readers who are interested in optimization but are not experts in NAS are not familiar with this.

It is unclear why latency and not the number of trainable parameters was chosen as the seconday objective for NAS-Bench-201. NAS-Bench-201 also reports the number of trainable parameters and latency is only briefly discussed in the paper.

The title is a bit misleading because this paper really focuses on multi-objective optimization and the method does not make sense in a single-objective scenario.

**Overall Review:**

*** Strong points ***

- For NAS, the approach is reasonable and shows good empirical performance.
- The motivation is clear.
- The experiments are well-described and seems to be complemented by detailed supplementary material.

*** Weak points ***

- In the experiments, there is a penality of 100 added to the validation error if a niche is empty. There is no argument for why this particular number was chosen. This choice should be motivated.
- The experiments only consider one secondary objective (number of trainable parameters), however, the general setup allows for an arbitrary number of secondary objectives. It would have been interesting to see the impact of more than one secondary objective.
- For the final experiment, training MobileNetV2, a comparison to other methods is missing.

**Potential Impact On The Field Of Automl:**

The paper addresses neural architecture search in a multi-objective setting. This is an important problem in cases where additional objectives should be optimized. The authors distinguish their method from more classical multi-objective optimization techniques and discuss advantages of their method.

As mentioned in their discussion, this method requires to define niches a-priori. If for example, it is unclear what memory is required to achieve a certain accuracy but memory is still an important consideration, this method would not be easily applicable compared to other multi-objective optimization techniques.

However, I believe that there exist plenty use-cases where the author's assumptions are met.

The paper focuses on neural-architecture search but the method has also potential impact on the broader black-box optimization community.

**Reproducibility:**

The reproducibility checklist is filled out sufficiently and the authors link to their code. I believe that it would be possible to reproduce the results once the appendix is available.

**Review Confidence:**

3: You are fairly confident in your assessment. It is possible that you did not understand some parts of the submission or that you are unfamiliar with some pieces of related work.

**Review Rating:**

5: Accept, good paper

**Review Summary:**

The paper presents an alternative way of incorporating secondary objectives in NAS by using quality-diversity optimization. They present a common use case, in particular the deployment of models on devices with different memory resources.  The paper is overall clear and well-motivated. The results show strong performance of their method. A comparison to other methods is missing on the final benchmark, also the choice of the secondary objective for NAS-Bench-201 was unclear to me.

Overall, I think some aspects need to be clarified but I overall recommend to accept the paper.

**Technical Quality And Correctness:**

The overall problem description is clear and it seems possible to reproduce the results based on the description in the paper (under some mild assumptions or once the appendix is published).

*** Improvements: ***

- The optimization problem of multi-objective NAS is formulated as min_{A\in \mathcal{A}} \bm{f}(A). This is generally not defined. While this notation is commonly used, it needs to be augmented by a definition of order (here, Pareto-dominance). The text reads as if the solution is generally given by the set of Pareto-optimal solutions but this depends on the choice of order.
- At the bottom of page 3, A^*_j is printed in boldface but nowhere else. This is confusing, I would remove the boldface here, it is not in boldface in Algorithm 1.

---

### Official Review · Reviewer_84bb · 2022-04-04

**Potential Impact On The Field Of Automl Rating:** 3
**Technical Quality And Correctness Rating:** 3
**Clarity:** Completely express the ideas to be co…
**Clarity Rating:** 4

**Summary Of Contributions:**

Quality-diversity optimization is a new branch of stochastic optimization, the author(s) has attempted to introduce QDO to the neural architecture search field. The paper framed the search for neuronal architecture as a QDO problem, and compared QDNAS and classic multi-object NAS whose performances are better. In the experimental results showed that QDNAS outperformed and more efficient multi-object NAS in the NAS-Bench-101 and NAS-Bench-201 with specified niches.

**Overall Review:**

Advantage:
The Quality-Diversity Optimization is a novel branch of stochastic optimization, the author(s) attempted to introduce QDO to neural architecture search field. Also, the experimental outcomes are sufficient, that a number of QDO and multi-object optimization methods have been tested on multiple benchmarks.

Key weakness:
The author did not provide code to carry out. Besides, it is regrettable that the difference in performance between QDO and multi-objects on various data sets has not been explored in depth.


**Potential Impact On The Field Of Automl:**

The QDO is still in its infancy. However, it offers a new perspective for NAS, there are still a lot of issues that need to be overcome, such as niche parameters. However, we believe that the introduction of QDO can provide NAS with greater opportunities to address the problems we face.

**Reproducibility:**

 The author did not provide code to carry out.

**Review Confidence:**

2: You are willing to defend your assessment, but it is quite likely that you did not understand the central parts of the submission or that you are unfamiliar with some pieces of related work.

**Review Rating:**

4: Marginally above the acceptance threshold (use sparsely)

**Review Summary:**

Overall, this paper is a comprehensive one, which clearly expresses the ideas to be conveyed and the experimental results are sufficient. While the experiment section has not delved into the details, this does not affect the excellence of this paper.

**Technical Quality And Correctness:**

The experimental outcomes are sufficient, that a number of QDO and multi-object optimization methods have been tested on multiple benchmarks. It is regrettable that the difference in performance between QDO and multi-objects on various data sets has not been explored in depth (the mean ranks or average performance was directly used as the conclusion). For the applicability of the algorithm to be possible, more experimentation is required to understand. Also, depending on the algorithm formula, QDO can actually be applied when k=1, that is, the test is in the single-domain state. I'm not sure why no relevant experiment has been given. After all, most NAS research is still in the single-domain.

---

### Official Review · Reviewer_L9w2 · 2022-04-06

**Potential Impact On The Field Of Automl Rating:** 2
**Technical Quality And Correctness Rating:** 2
**Clarity:** The paper is well structure and clear…
**Clarity Rating:** 3

**Summary Of Contributions:**

This paper introduces a new approach for multi-objective NAS that solves a quality diversity optimization (QDO) problem instead of the traditional pareto curve optimization problem.  The main insight presented by the authors is that we usually care about the best architecture in multiple perhaps overlapping subspaces instead of those on the pareto front.  The authors combine a few existing methods to form a multi-fidelity QDO solver called BOP-ElitesHB which combines Bayesian optimization with Hyperband for QDO.  Experiments show BOP-ElitesHB to outperform other methods for QDO on NASBench-101, NASBench201, and a model compression task.

**Overall Review:**

Positives:
- It is natural to pose NAS for multiple deployment scenarios as a QDO problem.
- The proprosed QDO optimizer BOP-EliteHB appears to work well on studied benchmarks.

Negatives:
- Technical novelty is limited since the main technical contribution is a small modification to Hyperband to adapt it for quality diversity optimization.  Then, the authors modify an existing QDO method called BOP-Elites to make it suitable for NAS search spaces by borrowing insights from SMAC and BANANAS.
- Benchmarks studied are not used in practice.  Need to evaluate QDO methods on a more realistic NAS search space like those based on MobileNetV3.

**Potential Impact On The Field Of Automl:**

The impact of the paper is limited because most experiments are conducted on tabular benchmarks with relatively small search spaces. The computational cost is also high due to the need to train individual architectures.  I think the impact would be higher if the authors included additional experiments comparing BOP-Elites to other search methods on pretrained NAS supernets like that in Once-for-all and BigNAS. This sort of approach would amortize the training cost over all niches and include more practical architectures in the search space.

**Reproducibility:**

I believe I would be able to reproduce the results with the information provided.

**Review Confidence:**

4: You are confident in your assessment, but not absolutely certain. It is unlikely, but not impossible, that you did not understand some parts of the submission or that you are unfamiliar with some pieces of related work.

**Review Rating:**

3: Marginally below the acceptance threshold (use sparsely)

**Review Summary:**

While I agree with QDO as the right problem to solve for multi-objective NAS, I think the paper would benefit from additional experiments on more practical NAS search spaces with niches corresponding to actual deployment settings (e.g. mobile platforms like Google Pixel, Samsung S7, etc).  In lieu of such experiments, the paper does not sufficiently establish QDO optimization as a meaningful approach to NAS in practice.

**Technical Quality And Correctness:**

In terms of methodology, the main technical contribution is a small modification to Hyperband to adapt it for quality diversity optimization.  Then, the authors modify an existing QDO method called BOP-Elites to make it suitable for NAS search spaces by borrowing insights from SMAC and BANANAS.  While straightforward, the proposed approaches are sound.

For the experiments, the authors compare to reasonable competitors and baselines and average over multiple seeds.  However, most of the results are on tabular benchmarks like NASBench101 and NASBench201.  The search space considered in NASBench201 is relatively small and we see BOP-ElitesHB outperform competitors on a majority of those benchmarks.  However, for NASBench101, which has a larger search space, ParEGOHB, a multi-objective method, is very competitive with QDO methods introduced by the author.  Practical NAS search spaces are more similar to that for NASBench101 so more evidence is needed.  I suggest adding experiments on the MobilNetV3 family of search spaces or on a pretrained supernet like that from Once-for-all.

---

### Official Review · Reviewer_5kWr · 2022-04-12

**Potential Impact On The Field Of Automl Rating:** 3
**Technical Quality And Correctness Rating:** 2
**Clarity Rating:** 3

**Summary Of Contributions:**

The authors demonstrate how multi-objective NAS can be formulated as a Quality Diversity Optimization (QDO).


**Clarity:**

The paper is clear with minor room for improvements. The paper can be improved during the rebuttal process.

**Overall Review:**

The authors demonstrate how multi-objective NAS can be formulated as a Quality Diversity Optimization (QDO). The formulation of the proposal is sound. Unfortunately, the authors don't demonstrate the idea of creating niches with actual hardware devices. It would have been great to show experiments on many target devices.



**Potential Impact On The Field Of Automl:**

The paper might have medium impact on the field but more experiments should be conducted.

**Reproducibility:**

Unfortunately, this paper has uploaded their code to Anonymous Github. The code is composed of hundreds of files that must be downloaded one by one. The code appears to be correct, but further testing is required.

**Review Confidence:**

4: You are confident in your assessment, but not absolutely certain. It is unlikely, but not impossible, that you did not understand some parts of the submission or that you are unfamiliar with some pieces of related work.

**Review Rating:**

4: Marginally above the acceptance threshold (use sparsely)

**Review Summary:**

The paper is marginally above the acceptance threshold. The quality of the paper is good but the experiments and results section could be improved, particularly when the authors show the creation of niches. It would have been good to see how the proposed approach would be used in actual hardware devices.

**Technical Quality And Correctness:**

The formulation of the proposal and pseudocode in the paper are sound, but the results are inconclusive. They create simple niches for compressed models using MobileNetV2 as baseline. A suggestion would be to run experiments on actual target devices to better demonstrate the creation of niches. Regarding resource usage, we suggest that the authors experiment with latency, memory requirement, and energy consumption.

---

### Meta-Review · Area_Chair_ZmNx · 2022-05-05

**Recommendation:** Accept
**Confidence:** 3

**Metareview:**

The paper proposes to consider solving a quality diversity optimization (QDO) problem instead of solving a traditional multi-objective optimization problem when performing hardware-aware NAS. The motivation is that in practice, one is typically more interested about finding the best configurations for different hardware (constraints) represented as different niches. Reviewers noted the pertinence of the problem and highlighted the quality of the results obtained. One of the main critic was to not consider actual hardware devices but this concern was addressed by the authors with extra experiments on MobileNet and two reviewers acknowledge this improvement and raise their score (one reviewer did not acknowledge this addition whereas it was one of his main critic).

Given the relevance of the problem studied and the good performance of the method proposed, I recommend accepting the paper.

A small comment, the following paper could be mentioned given the similarity with some aspect of the paper, in particular using multi-fidelity and a multi-objective approach for hardware-aware NAS:
A multi-objective perspective on jointly tuning hardware and hyperparameters (https://arxiv.org/abs/2106.05680).

---

### Decision · Program_Chairs · 2022-05-13

Accept